# Weakly supervised classification of aortic valve malformations using unlabeled cardiac MRI sequences

Jason A. Fries [1,2], Paroma Varma [3], Vincent S. Chen [1], Ke Xiao [4], Heliodoro Tejeda[4], Priyanka Saha[4], Jared Dunnmon[1], Henry Chubb[4], Shiraz Maskatia[4], Madalina Fiterau[1], Scott Delp[5], Euan Ashley [6,7,8], Christopher Ré[1,7,8] & James R. Priest [4,7,8]

Biomedical repositories such as the UK Biobank provide increasing access to prospectively collected cardiac imaging, however these data are unlabeled, which creates barriers to their use in supervised machine learning. We develop a weakly supervised deep learning model for classification of aortic valve malformations using up to 4,000 unlabeled cardiac MRI sequences. Instead of requiring highly curated training data, weak supervision relies on noisy heuristics defined by domain experts to programmatically generate large-scale, imperfect training labels. For aortic valve classification, models trained with imperfect labels substantially outperform a supervised model trained on hand-labeled MRIs. In an orthogonal validation experiment using health outcomes data, our model identifies individuals with a 1.8-fold increase in risk of a major adverse cardiac event. This work formalizes a deep learning baseline for aortic valve classification and outlines a general strategy for using weak supervision to train machine learning models using unlabeled medical images at scale.

[1] Department of Computer Science, Stanford University, Stanford, CA 94305, USA. [2] Center for Biomedical Informatics Research, Stanford University, Palo Alto, CA 94305, USA. [3] Department of Electrical Engineering, Stanford University, Palo Alto, CA 94305, USA. [4] Department of Pediatrics, Stanford University, Stanford, CA 94304, USA. [5] Department of Bioengineering, Stanford University, Palo Alto, CA 94305, USA. [6] Department of Medicine, Stanford University, Stanford, CA 94304, USA. [7] Chan Zuckerberg BioHub, San Francisco, CA 94158, USA. [8] These authors jointly supervised this work: Euan Ashley, Christopher Ré, James R. Priest. Correspondence and requests for materials should be addressed to J.A.F. (email: jason-fries@stanford.edu)

Aortic valve disease inclusive of bicuspid aortic valve (BAV) is the most common congenital malformation of the heart, occurring in 0.5–2% of the general population[1], and is associated with a variety of poor health outcomes[2]. In isolation, valvular dysfunction in BAV often leads to substantial cardiovascular pathology requiring surgical replacement of the aortic valve[3]. Machine learning models for automatically identifying aortic valve malformations via medical imaging could enable new insights into genetic and epidemiological associations with cardiac morphology. However, our understanding of the etiologies of BAV and its disease correlates have been limited by the variability in age of diagnosis and the absence of large, prospectively collected imaging datasets.

Obtaining labeled training data is one of the largest practical roadblocks to building machine learning models for use in medicine[4]. Recent deep learning efforts in medical imaging for detecting diabetic retinopathy[5] and cancerous skin lesions[6] each required more than 100,000 labeled images annotated by multiple physicians. Standard approaches to generating labeled data at scale such as crowdsourcing are poorly suited to medical images due to the domain expertise required and the logistics of working with protected health information. More fundamentally, labels are static artifacts with sunk costs: labels themselves do not transfer to different datasets and changes to annotation guidelines necessitate re-labeling data.

Recently, the UK Biobank released a dataset of >500,000 individuals with comprehensive medical record data prior to enrollment along with long-term followup. Importantly, these data also include prospectively obtained medical imaging and genome-wide genotyping data on 100,000 participants[7], including the first release of phase-contrast cardiac magnetic resonance imaging (MRI) sequences for 14,328 subjects. The high-dimensionality and overall complexity of these images make them appealing candidates for use with deep learning[8]. However, these prospectively collected MRIs are unlabeled, and the low prevalence of malformations such as aortic valve disease introduces considerable challenges in building labeled datasets at the scale required to train deep learning models.

In this work, we present a deep learning model for aortic valve malformation classification that is trained using largely unlabeled MRI data building on the paradigm of weak-supervision. Instead of requiring hand-labeled examples from cardiologists, we use new methods[9,10] to encode domain knowledge in the form of multiple, noisy heuristics or labeling functions which are applied to unlabeled data to generate imperfect training labels. This approach uses a factor graph-based model to estimate the unobserved accuracies of these labeling functions as well as infer statistical dependencies among labeling functions[11,12]. The resulting factor graph model is applied to unlabeled data to produce "de-noised" probabilistic labels, which are used to train a state-of-the-art hybrid Convolutional Neural Network/Long Short Term Memory (CNN-LSTM) model to classify aortic valve malformations. To assess the real-world relevance of our image classification model, we apply the CNN-LSTM to a cohort of 9230 new patients with long-term outcome and MRI data from the UK Biobank. In patients identified by our classifier as having BAV, we find a 1.8-fold increase in risk of a major adverse cardiac event. These findings demonstrate how weakly supervised methods help mitigate the lack of expert-labeled training data in cardiac imaging settings, and how real-world health outcomes can be learned directly from large-scale, unlabeled medical imaging data.

## Results

**Experiments.** We compare our weakly supervised models against two traditionally supervised baselines using identical CNN-LSTM architectures: (1) expert labels alone and (2) expert labels with data augmentation. Our supervised BASELINE model was trained using all hand-labeled MRIs from the development set. Due to class imbalance (6:100), training data was rebalanced by over-sampling BAV cases with replacement.

We evaluate the impact of training set size on weak supervision performance. These models are trained using only weakly labeled training data, i.e., no hand-labeled MRIs, built using a set of patients disjoint from our 412 gold annotation cohort. All probabilistic labels are split into positive and negative bins using a threshold of 0.5 and sampled uniformly at random with replacement to create balanced, training sets, e.g., sample 50 BAV and 50 tricuspid aortic valve (TAV) for a training set size of 100. We used balanced samples sizes of {50, 250, 500, 1000, 2000, 4000}. The final class balance for all 4239 weak labels in the training set was 264/3975 BAV/TAV. Full scale-up metrics for weak labels are shown in Fig. 1. Mean precision increased 128% (30.7 to 70.0) using 4239 weakly labeled MRIs; sensitivity (recall) matched performance of the expert-labeled baseline (53.3 vs. 60.0). At ≥1264 weak training examples, all models exceeded the performance of a model trained on 106 expert-labeled MRIs.

In Table 1, we report baseline model performance and the best weak supervision models found across all scale-up experiments. Models trained with 4239 weak labels and augmentation performed best overall, matching or exceeding all metrics compared to the best performing baseline model, expert labels with augmentation. The best weak supervision model had a 62% improvement in mean F1 score (37.8 to 61.4) and 128% higher mean precision (30.7 to 70.0). This model had higher mean area under the ROC curve (AUROC) (+13%) and normalized discounted cumulative gain (NDCG) (+57%) scores. See Supplementary Fig. 1 for ROC plots across all scale-up sizes.

Table 2 shows individual labeling function performance on test data, where metrics were computed per-frame. Precision, recall, and F1 scores were calculated by counting abstain votes as TAV labels, reflecting a strong prior on TAV cases. Individually, each function was a very weak classifier with poor precision (0–25.0) and recall (0–85.7), as well as mixed coverage (9.8–90%) and substantial conflict with other labeling functions (8–41.7%). Note that labeling functions provide both negative and positive class supervision, and sometimes performed best with a specific class, e.g., LF_Intensity targets negative (TAV) cases while LF_Perimeter targets positive (BAV) cases.

In total, 570/9230 subjects were classified as having BAV. In a time-to-event analysis encompassing up to 22 years of follow-up on the 9230 included participants with cardiac MRI data, the individuals with model-classified BAV showed a significantly lower MACE-free survival (hazard ratio 1.8; 95% confidence interval 1.3–2.4, $p = 8.83e−05$ log-rank test) (see Fig. 2) consistent with prior knowledge of co-incidence of BAV with comorbid cardiovascular disease[13,14]. In a linear model adjusted for age, sex, smoking, hyperlipidemia, diabetes, and BMI, individuals with model-classified BAV displayed a 2.5 mmHg increase in systolic blood pressure ($p < 0.001$).

Figure 3 shows a t-SNE plot of BAV/TAV clusters using the CNN-LSTM's last hidden layer output (i.e., the learned feature vector). In the post-hoc analysis of 36 predicted MRI labels, TAV cases had 94% (17/18) PPV (precision) and BAV cases had 61% (11/18) PPV, with BAV misclassifications occurring most often in cases with visible regurgitation and turbulent blood flow.

Table 3 shows the post-hoc analysis of 100 positive BAV predictions. In total, 28% of all positive predictions were true BAV cases, with 75% of predictions mapping to one or more valve pathologies of the aortic valve. Distribution across each sampled bucket (Q1–Q4) was largely uniform, indicating errors were randomly distributed in positive class predictions.

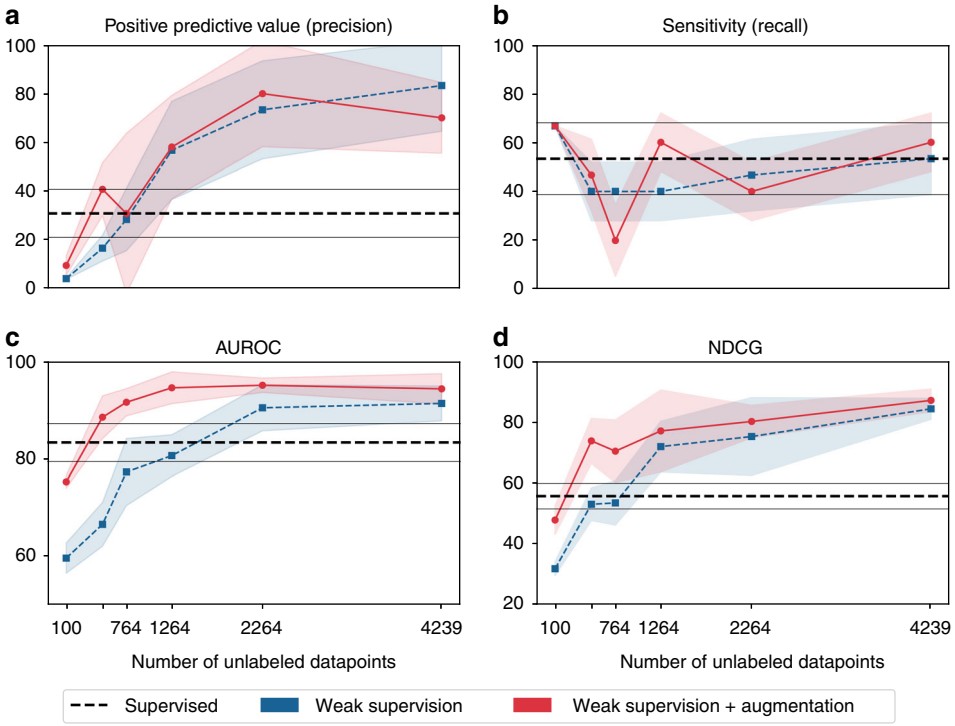

**Fig. 1** Weak supervision scale up performance metrics. Metrics include **a** positive predictive value (precision); **b** sensitivity (recall); **c** area under the ROC curve (AUROC); and **d** normalized discounted cumulative gain (NDCG). The y-axis is the score in [0,100] and the x-axis is the number of unlabeled MRIs used for training. The dashed horizontal line indicates the expert-labeled baseline model with augmentations. Shaded regions and gray horizontal lines indicate 95% confidence intervals (where $n$ = the number of unlabeled training MRIs)

---

**Table 1 Best weak supervision vs. hand labeled models**

| Model | Size | Precision | Recall | F1 | AUROC | NDCG |
|---|---|---|---|---|---|---|
| HL | 106 | 10.0 [1.3, 18.7] | 20.0 [5.4, 34.6] | 12.8 [2.5, 23.1] | 85.4 [80.8, 90.0] | 40.6 [36.4, 44.9] |
| HL + Aug. | 106 | 30.7 [20.8, 40.6] | 53.3 [38.7, 68.0] | 37.8 [27.7, 47.9] | 83.4 [79.5, 87.3] | 55.7 [51.5, 59.9] |
| WS | 4239 | **83.3** [64.5, 100.0] | 53.3 [38.7, 68.0] | 60.8 [50.6, 71.0] | 91.4 [87.8, 95.0] | 84.5 [81.1, 88.0] |
| WS + Aug. | 4239 | 70.0 [55.4, 84.6] | **60.0** [48.1, 72.0] | **61.4** [55.3, 67.5] | **94.4** [91.3, 97.6] | **87.3** [83.6, 91.0] |

WS indicates weak supervision models, HL indicates hand-labeled models, and Aug. indicates augmentation. Scores are computed with 95% confidence intervals (where $n$ = the size column), with bold text indicating best performance overall

---

**Table 2 Frame-level labeling function performance metrics**

| Labeling functions | Coverage (%) | Conflict (%) | Pos. Acc. | Neg. Acc. | Precision | Recall | F1 |
|---|---|---|---|---|---|---|---|
| LF_Area | 22.6 | 11.5 | 76.5 | 62.9 | 25.0 | 31.0 | 27.7 |
| LF_Perimeter | 9.8 | 8.0 | 100.0 | 0.0 | 20.8 | 26.2 | 23.2 |
| LF_Eccentricity | 87.4 | 38.9 | 85.7 | 42.3 | 12.7 | 85.7 | 22.1 |
| LF_Intensity | 28.9 | 24.1 | 0.0 | 69.0 | 0.0 | 0.0 | 0.0 |
| LF_Ratio | 90.4 | 41.7 | 67.5 | 49.6 | 10.7 | 64.3 | 18.3 |

---

## Discussion

In this work, we present a deep learning model for classifying aortic valve malformations from phase-contrast MRI sequences. These results were obtained using models requiring only a small amount of labeled data, combined with a large, imperfectly labeled training set generated via weak supervision. The success of this weak supervision paradigm, especially for a classification task with substantial class-imbalance such as BAV, is a step towards the larger goal of automatically labeling unstructured medical imaging from large datasets such as the UK Biobank. For medical

applications of machine learning as described here, we propose an additional standard of validation; that the model not only captures abnormal valve morphology, but more importantly the captured information is of real-world medical relevance and consistent with prior-knowledge of aortic valve pathology. Despite criteria selecting healthier individuals for study by MRI[15,16], individuals identified by our model showed more than an 1.8-fold increase in risk for comorbid cardiovascular disease.

Large unstructured medical imaging datasets are increasingly available to biomedical researchers, but the use of data on cardiac

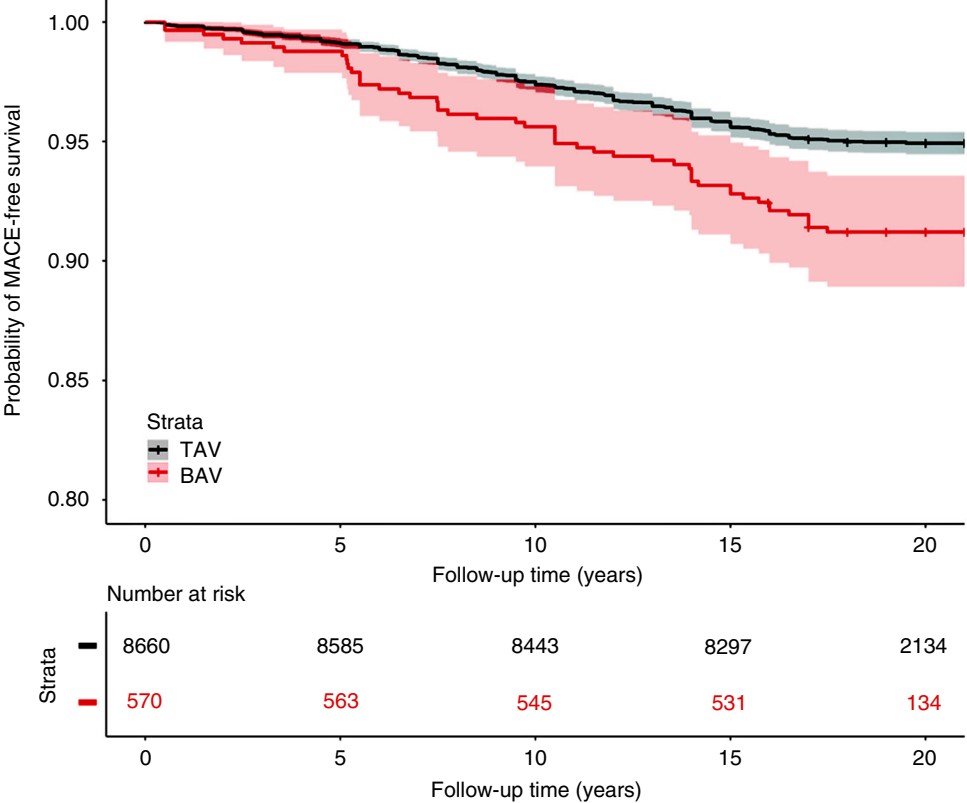

**Fig. 2** Unadjusted survival from MACE in 9230 participants stratified by model classification. MACE occurred in 59 of 570 individuals (10.4%) classified as BAV compared to 511 of 8660 individuals (5.9%) classified as TAV over the course of a median 19 years of follow up (hazard ratio 1.8; 95% confidence interval 1.3–2.4, $p = 8.83e-05$ log-rank test)

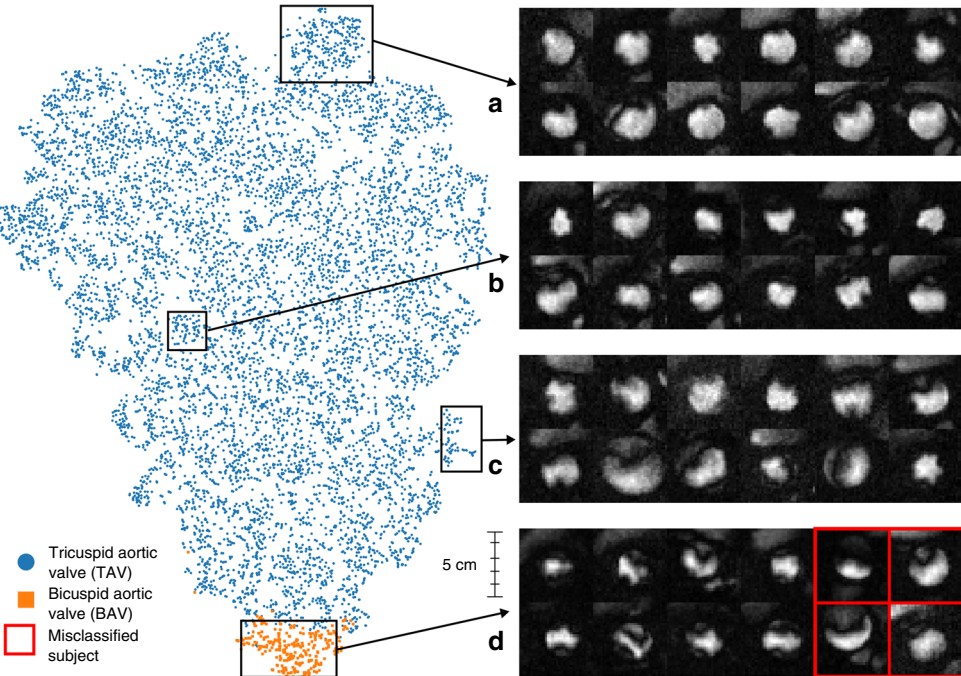

**Fig. 3** Patient clustering visualization. t-SNE visualization of the last hidden layer outputs of the CNN-LSTM model as applied to 9230 patient MRI sequences and **a–d** frames capturing peak flow through the aorta for a random sample of patients. Blue and orange dots represent TAV and BAV cases. The model clusters MRIs based on aortic shape and temporal dynamics captured by the LSTM. The top example box (**a**) contains clear TAV cases with very circular flow shapes, with (**b**) and (**c**) becoming more irregular in shape until (**d**) shows highly irregular flow typical of BAV. Misclassifications of BAV (red boxes) generally occur when the model fails to differentiate regurgitation of the aortic valve and turbulent blood flow through a normal appearing aortic valve orifice

**Table 3 Prediction set validation**

|  | Q1 | Q2 | Q3 | Q4 | Overall |
|---|---|---|---|---|---|
| **Total BAV** | 24% (6) | 28% (7) | 36% (9) | 24% (6) | 28% |
| **Total non-BAV valve pathologies** | 48% (12) | 44% (11) | 40% (10) | 56% (14) | 47% |
| **Total flow/image artifacts** | 28% (7) | 28% (7) | 24% (6) | 20% (5) | 25% |
| *Aortic stenosis* | 40% (10) | 44% (11) | 28% (7) | 36% (9) | 37% |
| *Aortic insufficiency* | 4% (1) | 8% (2) | 16% (4) | 16% (4) | 11% |
| *Tethered/thickened leaflet* | 16% (4) | 4% (1) | 20% (5) | 24% (6) | 16% |
| *Turbulent flow artifact* | 40% (10) | 40% (10) | 20% (5) | 36% (9) | 34% |
| *Image artifact* | 4% (1) | 4% (1) | 0% (0) | 4% (1) | 3% |
| Total Subjects | 25 | 25 | 25 | 25 | 100 |

Bold rows are disjoint category counts for true BAV, confounding non-BAV valve pathologies, and imaging artifacts. Italicized rows contain categories where counts may overlap with non-BAV valve pathologies and image artifacts

morphology derived from medical imaging depends upon their integration into genetic and epidemiological studies. For most aspects of cardiac structure and function, the computational tools used to perform clinical measurements require the input or supervision of an experienced user, typically a cardiologist, radiologist, or technician. Large datasets exploring cardiovascular health such as MESA and GenTAC which both include imaging data have been limited by the scarcity of expert clinical input in labeling and extracting relevant information[17,18]. Our approach provides a scalable method to accurately and automatically label such high value datasets.

Automated classification of imaging data represents the future of imaging research. Weakly supervised deep learning tools may allow imaging datasets from different institutions which have been interpreted by different clinicians, to be uniformly ascertained, combined, and analyzed at unprecedented scale, a process termed harmonization. Independent of any specific research or clinical application, new machine learning tools for analyzing and harmonizing imaging data collected for different purposes will be the critical link that enables large-scale studies to connect anatomical and phenotypic data to genomic information, and health-related outcomes. For the purposes of research, such as genome-wide association studies, higher precision (positive predictive value) is more important for identifying cases. Conversely, in a clinical application, the flagging of all possible cases of malformations for manual review by a clinician would be facilitated by a more sensitive threshold. The model presented here can be tuned to target either application setting.

Our analytical framework and models have limitations. Estimation of the true prevalence of uncommon conditions such as BAV and ascertainment of outcomes within a given population is complicated by classical biases in population health science. Registries of BAV typically enroll patients only with clinically apparent manifestations or treatment for disease which may not account for patients who do not come to medical attention.

Estimates derived from population-based surveillance are usually limited to relatively small numbers of participants due to the cost and difficulty of prospective imaging, and small cohort sizes impede accurate estimates for rare conditions such as BAV. Age and predisposition to research participation may also affect estimates of disease prevalence, a documented phenomenon within the UK Biobank[19]. Morbidity and mortality from BAV are accrued cumulatively over time, thus studies of older participants are missing individuals with severe disease who may have died or been unable to participate. Conversely calcific aortic valve disease, which increases in incidence with age, may result in an acquired form of aortic stenosis difficult to distinguish from BAV by cardiac flow imaging[20].

A structured post-hoc analysis of 100 model-classified aortic valve malformations showed that the model is broadly sensitive to the detection of aortic valve pathology including BAV, but also aortic stenosis, aortic insufficiency, and the presence of thickened or tethered aortic valve leaflets (Table 3). Relative to a normally functioning aortic valve with a circular or symmetrically triangular appearing pattern of flow, each of these pathologies may result in turbulent blood flow which appears asymmetric or non-uniform in phase-contrast imaging of the aortic valve (Fig. 3). Thus even for the current best-performing model, one displaying good predictive characteristics for a class-imbalanced problem, misclassification events do occur. However, many of these failure modes are challenging even for clinicians to discriminate when restricted to the single MRI view utilized in this study. Integrating additional views of the aorta can help clinicians discriminate BAV from these other valve pathologies, underlining the need to explore machine learning models that synthesize multiple streams of MRI data. Incorporating side information from ICD9/10 and OPCS-4 codes to leverage data on long-term outcomes and confounding pathologies is another exciting area for future model improvement.

This work demonstrates how weak supervision can be used to train a state-of-the-art deep learning model for aortic valve malformation classification using unlabeled MRI sequences. Using domain heuristics encoded as functions to programmatically generate large-scale, imperfect training data provided substantial improvements in classification performance over models trained on hand-labeled data alone. Transforming domain insights into labeling functions instead of static labels mitigates some of the challenges inherent in the domain of medical imaging, such as extreme class imbalance, limited training data, and scarcity of expert input. Most importantly, our BAV classifier successfully identified individuals at long-term risk for cardiovascular disease, demonstrating real-world relevance of imaging models built using weak supervision techniques.

## Methods

**Dataset**. From 2006 to 2010, the UK Biobank recruited 502,638 participants aged 37–73 years in an effort to create a comprehensive, publicly available health-targeted dataset. The initial release of UK Biobank imaging data includes cardiac MRI sequences for 14,328 subjects[21], including eight cardiac imaging sets. Three sequences of phase-contrast MRI images of the aortic valve registered in an en face view at the sinotubular junction were obtained. Figure 4 shows example MRI videos in all encodings: raw anatomical images (CINE); magnitude (MAG); and velocity encoded (VENC)[22]. See Supplementary Movies 1–6 for video examples. In MAG and VENC series, pixel intensity directly maps to velocity of blood flow. This is performed by exploiting the difference in phase of the transverse magnetism of protons within blood when flowing parallel to a gradient magnetic field, where the phase difference is proportional to velocity. CINE images encode anatomical information without capturing blood flow. All raw phase contrast MRI sequences are 30 frames, 12-bit grayscale color, and 192 × 192 pixels.

Studies using the UK Biobank are exempt from approval by the Stanford University School of Medicine Institutional Review Board as the data is de-identified and publicly available. Informed consent for use of health information and imaging was performed by the UK Biobank organization at the time of

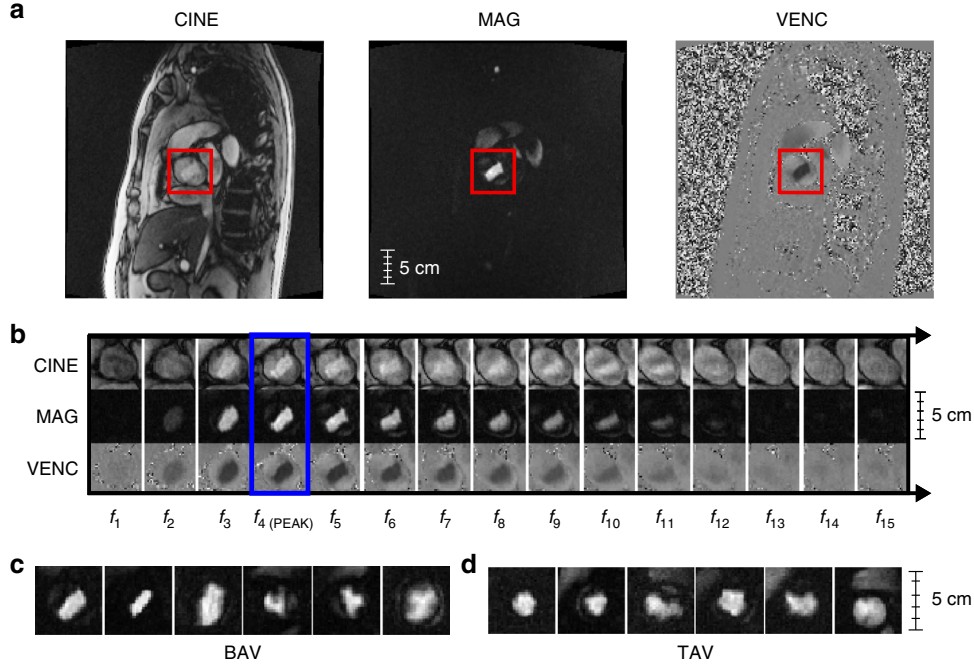

**Fig. 4** Example MRI sequence data for BAV and TAV subjects. **a** Uncropped MRI frames for CINE, MAG, and VENC series in an oblique coronal view of the thorax centered upon an en face view of the aortic valve at sinotubular junction (red boxes). **b** 15-frame subsequence of a phase-contrast MRI for all series, with peak frame outlined in blue. MAG frames at peak flow for 12 patients, broken down by class: **c** bicuspid aortic valve (BAV) and **d** tricuspid aortic valve (TAV)

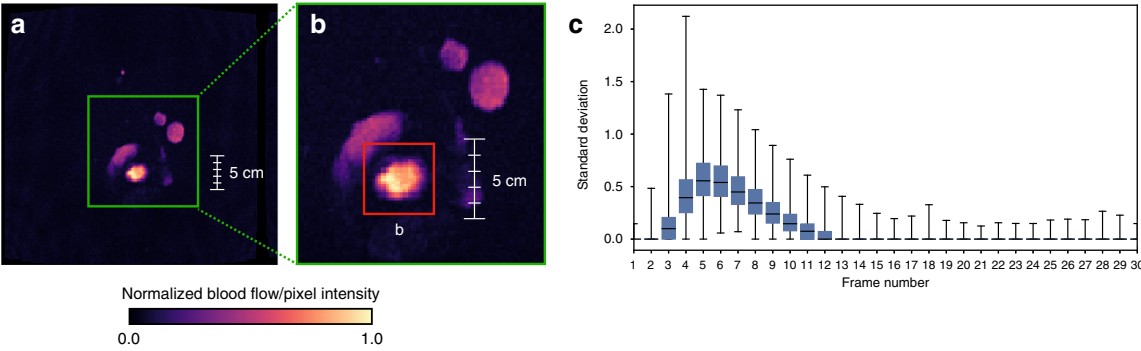

**Fig. 5** Aorta localization. **a** Uncropped MAG series MRI frame, showing 0–1 normalized, per pixel standard deviation. **b** Green box is a zoom of the heart region and the red box corresponds to the aorta—the highest weighted pixel area in the image. **c** A box and whisker plot of per-frame standard deviations for all 4239 MRI sequences in the weak training set. Here the blue box represents the interquartile range of the first and third quartiles, the black line is the median value, and the whiskers map to the minimum and maximum values across all frames at a given index. Note the most variation occurs in the first 15 frames

participant enrollment. The UK Biobank ethics committee administered the consent and regulatory compliance for all research participants[23]. The collection, distribution, and use of UK Biobank data for non-commercial research purposes is compliant with all relevant regulations including European Union General Data Protection Regulation.

**MRI preprocessing**. All MRIs were preprocessed to: (1) localize the aortic valve to a $32 \times 32$ crop image size; and (2) align all image frames by peak blood flow in the cardiac cycle. Since the MAG series directly captures blood flow—and the aorta typically has the most blood flow—both of these steps are straightforward using standard threshold-based image processing techniques when the series is localized to a cross-sectional plane at the sinotubular junction. Selecting the pixel region with maximum standard deviation across all frames localized the aorta, and selecting the frame with maximum standard deviation identified peak blood flow. See Fig. 5 and Supplementary Methods for implementation details. Both heuristics were very accurate (>95% as evaluated on the development set) and selecting a ±7 frame window around the peak frame $f_{peak}$ captured 99.5% of all pixel variation for the aorta. All three MRI sequences were aligned to this peak before classification.

**Gold standard annotations**. Gold standard labels were created for 412 patients (12,360 individual MRI frames), with each patient labeled as BAV or TAV, i.e., having two vs. the normal three aortic valve leaflets. We focus our analysis on BAV as it is the easiest malformation to identify from this MRI view. Total annotations included: a development set (100 TAV and 6 BAV patients) for writing labeling functions; a validation set (208 TAV and 8 BAV patients) for model hyperparameter tuning; and a held-out test set (87 TAV and 3 BAV patients) for final evaluation. The development set was selected via chart review of administrative codes (ICD9, ICD10, or OPCS-4) consistent with BAV and followed by manual annotation. The validation and test sets were sampled at random with uniform probability from all 14,328 MRI sequences to capture the BAV class distribution expected at test time. Development and validation set MRIs were annotated by a single cardiologist (J.R.P.). All test set MRIs were annotated by 3 cardiologists (J.R.P., H.C., S.M.) and final labels were assigned based on a majority vote across annotators. For inter-annotator agreement on the test set, Fleiss's Kappa statistic was 0.354. This reflects a fair level of agreement amongst annotators given the difficulty of the task[24,25]. Test data was withheld during all aspects of model development and used solely for the final model evaluation.

**Weak supervision**. There is considerable research on using indirect or weak supervision to train machine learning models for image and natural language tasks without relying entirely on manually labeled data[9,26,27]. One longstanding approach is distant supervision[28,29], where indirect sources of labels are used to generate noisy training instances from unlabeled data. For example, in the ChestX-ray8 dataset[30] disorder labels were extracted from clinical assessments found in radiology reports. Unfortunately, we often lack access to indirect labeling resources or, as in the case of BAV, the class of interest itself may be rare and underdiagnosed in existing medical records. Another strategy is to generate noisy labels via crowdsourcing[31,32], which in some medical imaging tasks can perform as well as trained experts[33,34]. In practice, however, crowdsourcing is logistically difficult when working with protected health information such as MRIs. A significant challenge in all weakly supervised approaches is correcting for label noise, which can negatively impact end model performance. Noise is commonly addressed using rule-based and generative modeling strategies for estimating the accuracy of label sources[35,36].

In this work, we use the recently proposed data programming[9] method, a generalization of distant supervision that uses a factor graph-based model to learn both the unobserved accuracies of labeling sources and statistical dependencies between those sources[11,12]. In this approach, source accuracy and dependencies are estimated without requiring labeled data, enabling the use of weaker forms of supervision to generate training data, such as noisy heuristics from clinical experts. For example, in BAV patients the phase-contrast imaging of flow through the aortic valve has a distinct ellipse or asymmetrical triangle appearance compared to the more circular aorta in TAV patients. This is the reasoning a human might apply when directly examining an MRI. In data programming these types of broad, often imperfect domain insights are encoded into functions that vote on the potential class label of unlabeled data points. This allows us to weakly supervise tasks where indirect label sources, such as patient notes with assessments of BAV, are not available.

The idea of encoding domain insights is formalized as labeling functions—black box functions which vote on unlabeled data points. Labeling function output is used to learn a probabilistic label model of the underlying annotation process, where each labeling function is weighted by its estimated accuracy to generate probabilistic training labels $y_i \in [0, 1]$. These probabilistically labeled data are then used to train an off-the-shelf discriminative model such as a deep neural network. The only restriction on labeling functions is that they vote correctly with probability better than random chance. In images, labeling functions are defined over a set of domain features or primitives, semantic abstractions over raw pixel data that enable experts to more easily encode domain heuristics. Primitives encompass a wide range of abstractions, from simple shape features to complex semantic objects such as anatomical segmentation masks. Critically, the final discriminative model learns features from the original MRI sequence, rather than the restricted space of primitives used by labeling functions. This allows the model to generalize beyond the heuristics encoded in labeling functions.

Patient MRIs are represented as a collection of $m$ frames $\mathbf{X} = \{x_1, …, x_m\}$, where each frame $x_i$ is a $32 \times 32$ image with MAG, CINE, and VENC encodings mapped to color channels. Each frame is modeled as an unlabeled data point $x_i$ and latent random variable $y_i \in [-1, 1]$, corresponding to the true (unobserved) frame label. Supervision is provided as a set of $n$ labeling functions $\lambda_1, …, \lambda_n$ that define a mapping $\lambda_j : x_i \rightarrow \Lambda_{ij}$ where $\Lambda_{i1}, …, \Lambda_{in}$ is the vector of labeling function votes for sample $i$. In binary classification, $\Lambda_{ij}$ is in the domain $\{-1, 0, 1\}$, i.e., {false, abstain, true}, resulting in a label matrix $\Lambda \in \{-1, 0, 1\}^{m \times n}$.

The relationship between unobserved labels $y$ and the label matrix $\Lambda$ is modeled using a factor graph[37]. We learn a probabilistic model that best explains $\Lambda$, i.e., the matrix observed by applying labeling functions to unlabeled data. When labeling function outputs are conditionally independent given the true label, this model consists of $n$ accuracy factors between $\lambda_1, …, \lambda_n$ and $y$

$$\phi_j^{Acc}(\Lambda_i, y_i) := y_i \Lambda_{ij} \tag{1}$$

$$p_\theta(\Lambda, \mathbf{Y}) \propto \exp\left(\sum_{i=1}^{m} \sum_{j=1}^{n} \theta_j^{Acc} \phi_j^{Acc}(\Lambda_i, y_i)\right) \tag{2}$$

where $\mathbf{Y} := y_i, …, y_m$, our true labels. The model's weights $\theta$ are estimated by minimizing the negative log likelihood of $p_\theta(\Lambda)$ using contrastive divergence[38]. Optimization is done using standard stochastic gradient descent with Gibbs sampling for gradient estimation.

In many settings, we encounter statistical dependencies among labeling functions. These dependencies are included in the model by defining additional factors

$$p_\theta(\Lambda, \mathbf{Y}) \propto \exp\left(\sum_{i=1}^{m} \sum_{t \in T} \sum_{s \in S_t} \theta_s^t \phi_s^t(\Lambda_i, y_i)\right) \tag{3}$$

where $t \in T$ is a dependency type and $S_t$ are the labeling functions that participate in $t$. These dependencies may be specified manually if known or learned from unlabeled data.

Automatically learning dependencies from unlabeled data is important in weakly supervised imaging tasks where labeling functions interact with a small set of primitives and have higher order dependency structure. For example, a labeling

function defined using the ratio of area and perimeter has dependencies with separate labeling functions for area and perimeter. By expressing supervision using a small space of primitives, we can rely on the Coral method[11] to statically analyze labeling function source code and automatically infer complex dependencies among labeling functions based on which primitives they use as input.

The final weak supervision pipeline requires two inputs: (1) primitive feature matrix; and (2) observed label matrix $\Lambda$. For generating $\Lambda$, we take each patient's frame sequence $\bar{x}_i = \{x_{1i}, …x_{30i}\}$ and apply labeling functions to a window of $t$ frames $\{x_{(f_{peak}-t/2)i}, …, x_{(f_{peak}+t/2)i}\}$ centered on $f_{peak}$, i.e., the frame mapping to peak blood flow. Here $t = 6$ performed best in our label model experiments. The output of the label model is a set of per frame probabilistic labels $\{y_1, …, y_{(t \times N)}\}$ where $N$ is the number of patients. To compute a single, per patient probabilistic label, $\bar{y}_i$, we assign the mean probability of all $t$ patient frames if mean($\{y_{1i}, …, y_{ti}\}$) > 0.9 and the minimum probability if min($\{y_{1i}, …, y_{ti}\}$) < 0.5. Patient MRIs that did not meet these thresholds (7%; 304/4543) were removed from the final weak label set. The final weakly labeled training set consists of each 30 frame MRI sequence and a single probabilistic label per-patient: $\hat{\mathbf{X}} = \{\bar{x}_1, …, \bar{x}_N\}$ and $\hat{\mathbf{Y}} = \{\bar{y}_1, …, \bar{y}_N\}$.

Primitives are generated using existing models or methods for extracting features from image data. In our setting, we restricted primitives to unsupervised shape statistics and pixel intensity features provided by off-the-shelf image analysis tools such as scikit-image[39]. Primitives are generated using a binarized mask of the aortic valve for each frame in a patient's MAG series. Since the label model accounts for noise in labeling functions and primitives, we can use unsupervised thresholding techniques such as Otsu's method[40] to generate binary masks. All masks were used to compute primitives for: (1) area; (2) perimeter; (3) eccentricity (a [0,1) measure comparing the mask shape to an ellipse, where 0 indicates a perfect circle); (4) pixel intensity (the mean pixel value for the entire mask); and (5) ratio (the ratio of area over perimeter squared). Since the size of the heart and anatomical structures correlate strongly with patient sex, we normalized these features by two population means stratified by sex in the unlabeled set.

We designed 5 labeling functions using the primitives described above. For model simplicity, labeling functions were restricted to using threshold-based, frame-level information for voting. All labeling function thresholds were selected manually using distributional statistics computed over all primitives for the expert-labeled development set. See Supplementary Fig. 2 for the complete development set used for labeling function design and Supplementary Table 1 for labeling function implementations. The final weak supervision pipeline is shown in Fig. 6.

The discriminative model classifies BAV/TAV status using $t$ contiguous MRI frames ($5 \le t \le 30$, where $t$ is a hyperparameter) and a single probabilistic label per patient. This model consists of two components: a frame encoder for learning frame-level features and a sequence encoder for combining individual frames into a single feature vector. For the frame encoder, we use a Dense Convolutional Network (DenseNet)[41] with 40 layers and a growth rate of 12, pretrained on 50,000 images from CIFAR-10[42]. We tested two other common pretrained image neural networks (VGG16[43], ResNet-50[44]), but found that a DenseNet40-12 model performed best overall, aligning with the previous reports[41]. The DenseNet architecture takes advantage of low-level feature maps at all layers, making it well-suited for medical imaging applications where low-level features (e.g., edge detectors) often carry substantial explanatory power.

For the sequence encoder, we used a Bidirectional Long Short-term Memory (LSTM)[45] sequence model with soft attention[46] to combine all MRI frame features. The soft attention layer optimizes the weighted mean of frame features, allowing the network to automatically give more weight to the most informative frames in an MRI sequence. We explored simpler feature pooling architectures (e.g., mean/max pooling), but each of these methods was outperformed by the LSTM in our experiments. The final hybrid CNN-LSTM architecture aligns with recent methods for state-of-the-art video classification[47,48] and 3D medical imaging[49].

The CNN-LSTM model is trained using noise-aware binary cross entropy loss $L$:

$$\hat{w} = argmin_w \frac{1}{N} \sum_{i=1}^{N} \mathbb{E}_{y \sim \hat{Y}}[L(w, x_i, y)] \tag{4}$$

This is analogous to standard supervised learning loss, except we are now minimizing the expected value with respect to $\hat{Y}$[9]. This loss enables the discriminative model to take advantage of the more informative probabilistic labels produced by the label model, i.e., training instances with higher probability have more impact on the learned model. Figure 7 shows the complete discriminative model pipeline.

**Training and hyperparameter tuning**. The development set was used to write all labeling functions and the validation set was used for all model hyperparameter tuning. All models were evaluated with and without data augmentation. Data augmentation is used in deep learning models to increase training set sizes and encode known invariances into the final model by creating transformed copies of existing samples. For example, BAV/TAV status does not change under translation, so generating additional shifted MRI training images does not change the class label, but does improve final classification performance. We used a combination of crops and affine transformations commonly used by state-of-the-art image classifiers[50]. We tested models using all 3 MRI series (CINE, MAG, VENC with a

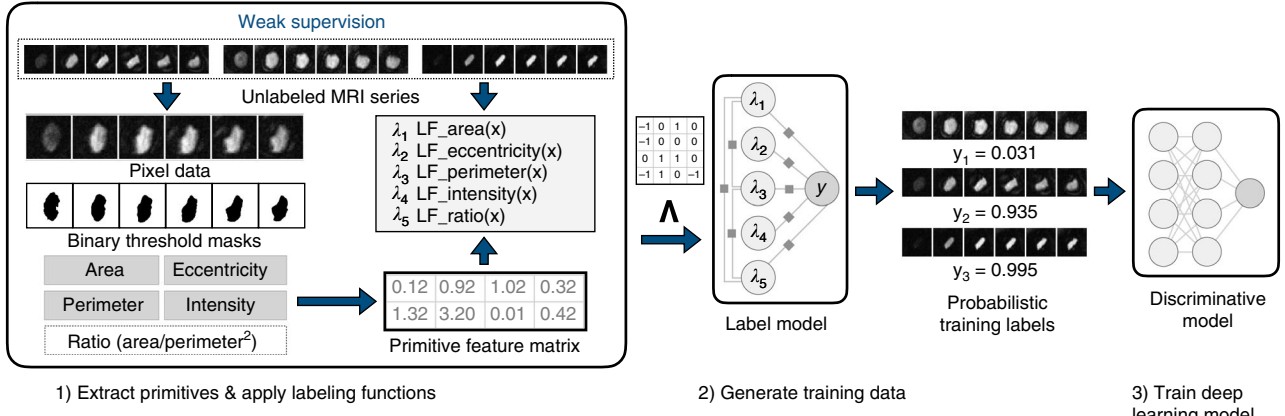

**Fig. 6** Weak supervision workflow. Pipeline for probabilistic training label generation based on user-defined primitives and labeling functions. Primitives and labeling functions (step 1) are used to weakly supervise the BAV classification task and programmatically generate probabilistic training data from large collections of unlabeled MRI sequences (step 2), which are then used to train a noise-aware deep learning classification model (step 3)

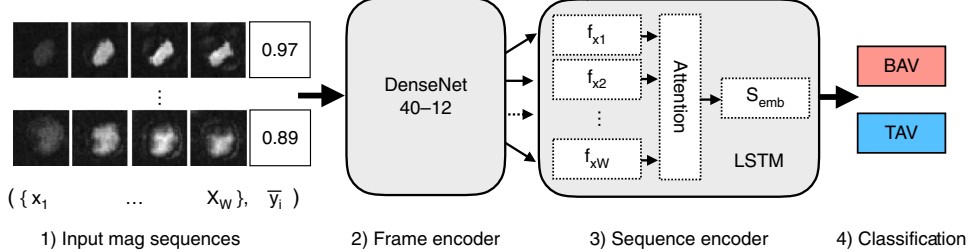

**Fig. 7** Deep neural network for MRI sequence classification. Each MRI frame is encoded by the DenseNet into a feature vector $f_{xi}$. These frame features are fed in sequentially to the LSTM sequence encoder, which uses a soft attention layer to learn a weighted mean embedding of all frames $S_{emb}$. This forms the final feature vector used for binary classification

single series per channel) and models using only the MAG series. The MAG series performed best, so we only report those results here.

Hyperparameters were tuned for L2 penalty, dropout, learning rate, and the feature vector size of our last hidden layer before classification. Augmentation hyperparameters were tuned to determine final translation, rotation, and scaling ranges. All models use validation-based early stopping with F1 score as the stopping criterion. The probability threshold for classification was tuned using the validation set for each run to address known calibration issues when using deep learning models[51]. Architectures were tuned using a random grid search over 10 models for non-augmented data and 24 for augmented data. See Supplementary Table 2 for full parameter grid settings.

**Evaluation metrics**. Classification models were evaluated using positive predictive value (precision), sensitivity (recall), F1 score (i.e., the harmonic mean of precision and recall), and AUROC. Due to the extreme class imbalance of this task we also report discounted cumulative gain (DCG) to capture the overall ranking quality of model predictions[52]. Each BAV or TAV case was assigned a relevance weight $r$ of 1 or 0, respectively, and test set patients were ranked by their predicted probabilities. DCG is computed as $\sum_i^p \frac{r_i}{\log_2(i+1)}$ where $p$ is the total number of instances and $i$ is the corresponding rank. This score is normalized by the DCG score of a perfect ranking (i.e., all true BAV cases in the top ranked results) to compute normalized DCG (NDCG) in the range [0.0, 1.0]. Higher NDCG scores indicate that the model does a better job of ranking BAV cases higher than TAV cases. All scores were computed using test set data, using the best performing models found during grid search, and reported as the mean and 95% confidence intervals of 5 different random model weight initializations.

For labeling functions, we report two additional metrics: coverage (Eq. (5)) a measure of how many data points a labeling function votes {−1, 1} on; and conflict (Eq. (6)) the percentage of data points where a labeling function disagrees with one or more other labeling functions.

$$\text{coverage}_{\lambda_j} = \frac{1}{N}\sum_{i=1}^{N} 1\left(\lambda_j(x_i) \in \{-1, 1\}\right) \qquad (5)$$

$$\text{conflict}_{\lambda_j} = \frac{1}{N}\sum_{i=1}^{N} 1\left(\sum_{k \neq j}^{\lambda_n} 1\left(\lambda_j(x_i) \in \{-1, 1\} \wedge \lambda_j(x_i) \neq \lambda_k(x_i)\right)\right) > 0 \qquad (6)$$

**Model evaluation with clinical outcomes data**. To construct a real-world validation strategy dependent upon the accuracy of image classification but completely independent of the imaging data input, we used model-derived classifications (TAV vs. BAV) as a predictor of validated cardiovascular outcomes using standard epidemiological methods. For 9230 patients with prospectively obtained MRI imaging who were excluded from any aspect of model construction, validation, or testing we performed an ensemble classification with the best performing CNN-LSTM model.

For evaluation we assembled a standard composite outcome of major adverse cardiovascular events (MACE). Phenotypes for MACE were inclusive of the first occurrence of coronary artery disease (myocardial infarction, percutaneous coronary intervention, coronary artery bypass grafting), ischemic stroke (inclusive of transient ischemic attack), heart failure, or atrial fibrillation. These were defined using ICD-9, ICD-10, and OPCS-4 codes from available hospital encounter, death registry, and self-reported survey data of all 500,000 participants of the UK Biobank at enrollment similar to previously reported methods[53].

Starting 10 years prior to enrollment in the study, median follow up time for the participants with MRI data included in the analysis was 19 years with a maximum of 22 years. For survival analysis, we employed the "survival" and "survminer" packages in R version 3.4.3, using aortic valve classification as the predictor and time-to-MACE as the outcome, with model evaluation by a simple log-rank test.

To verify the accuracy of the CNN-LSTM's predicted labels, we generated 2 subsets of our model's predictions for manual review: (1) 36 randomly chosen MRI sequences (18 TAV and 18 BAV patients); and (3) 100 positive BAV predictions, binned into quartiles by predicted probability. All MRIs were reviewed and labeled by a single annotator (J.R.P.). The output of the last hidden layer was visualized using a t-distributed stochastic neighbor embedding (t-SNE)[54] plot to assist error analysis.

**Related work**. In medical imaging, weak supervision refers to a broad range of techniques using limited, indirect, or noisy labels. Multiple instance learning (MIL) is one common weak supervision approach in medical images[55]. MIL approaches

assume a label is defined over a bag of unlabeled instances, such as an image-level label being used to supervise a segmentation task. Xu et al.[56] simultaneously performed binary classification and segmentation for histopathology images using a variant of MIL, where image-level labels are used to supervise both image classification and a segmentation subtask. ChestX-ray8[30] was used in Li et al.[57] to jointly perform classification and localization using a small number of weakly labeled examples. Patient radiology reports and other medical record data are frequently used to generate noisy labels for imaging tasks[30,58–60].

Weak supervision shares similarities with semi-supervised learning[61], which enables training models using a small labeled dataset combined with large, unlabeled data. The primary difference is how the structure of unlabeled data is specified in the model. In semi-supervised learning, we make smoothness assumptions and extract insights on structure directly from unlabeled data using task-agnostic properties such as distance metrics and entropy constraints[62]. Weak supervision, in contrast, relies on directly injecting domain knowledge into the model to incorporate the underlying structure of unlabeled data. In many cases, these sources of domain knowledge are readily available in existing knowledge bases, indirectly-labeled data like patient notes, or weak classification models and heuristics.

**Reporting summary**. Further information on research design is available in the Nature Research Reporting Summary linked to this article.

## Data availability
All primary data that support the findings of this study are publicly available from the UK Biobank organization by application for academic non-commercial use: https://www.ukbiobank.ac.uk/using-the-resource/. All remaining data contained in the manuscript will be made available from the corresponding author upon reasonable request.

## Code availability
All code used in this study was written in Python v2.7. Deep learning models were implemented using PyTorch v3.1. Preprocessing code, deep learning implementations, experimental scripts, and trained BAV classifications models are all open source and available at: https://github.com/HazyResearch/ukb-cardiac-mri; https://doi.org/10.5281/zenodo.2654330.

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

## Acknowledgements

The authors thank Seung-Pyo Lee for his initial contributions to the start of this project. The authors thank the Brin-Wojcicki Foundation for support (E.A., J.R.P.). This work was supported in part by the Mobilize Center, a National Institutes of Health Big Data to Knowledge (BD2K) Center of Excellence supported through Grant U54EB020405 (J.A.F., M.F.), the National Science Foundation (NSF) Graduate Research Fellowship under No. DGE-114747 (P.V.), Joseph W. and Hon Mai Goodman Stanford Graduate Fellowship (P.V.), and the Intelligence Community Postdoctoral Fellowship (J.D.). The authors gratefully acknowledge the support of the Defense Advanced Research Projects Agency (DARPA) SIMPLEX program under No. N66001-15-C-4043, DARPA FA8750-12-2-0335, and FA8750-13-2-0039, DOE 108845, National Institute of Health (NIH) U54EB020405, the Office of Naval Research (ONR) under award Nos. N000141210041 and N000141310129, the Moore Foundation, Annenberg Graduate Fellowship, the Okawa Research Grant, American Family Insurance, Accenture, Toshiba, and Intel. This material is based on research sponsored by DARPA under agreement number FA8750-17-2-0095. The U.S. Government is authorized to reproduce and distribute reprints for Governmental purposes notwithstanding any copyright notation thereon. This research was supported in part by affiliate members and other supporters of the Stanford DAWN project: Intel, Microsoft, Teradata, and VMware. Any opinions, findings, and conclusions or recommendations expressed in this material are those of the authors and do not necessarily reflect the views of DARPA, AFRL, NSF, NIH, ONR, or the U.S. Government. The views and conclusions contained herein are those of the authors and should not be interpreted as necessarily representing the official policies or endorsements, either expressed or implied, of DARPA or the U.S. Government.

## Author contributions

J.R.P. conceived the initial study. J.A.F., P.V., V.S.C., K.X., H.T., and J.D. wrote code and conducted experimental analysis of machine learning models. J.R.P., H.C., and S.M. annotated validation data. H.T., K.X., and P.S. handled data preprocessing. P.S. contributed the survival analysis models. J.A.F., P.V., M.F., S.D., E.A., C.R., and J.R.P. contributed ideas and experimental designs. All authors contributed to writing.

## Additional information

**Competing interests:** The authors declare no competing interests.

