## [Peer Review File · Nature Communications]

Reviewers' comments:

Reviewer #1 (Remarks to the Author):

Identification and characterization of bicuspid aortic valve anatomy is a human-intensive task with considerable inter-observer variability and about a 5-10% failure to accurately define the valve. The problem is notable for the several anatomical variants and difficulty identifying partial commissural fusion. The problem is especially difficult using gated raw-image cardiac MRI, but the value of avoiding ionizing radiation in long-term management of these patients is important. These imaging problems are a hindrance to clinical diagnosis, high-quality epidemiologic studies and accurate phenotyping for high-N genetic studies.

The authors present a deep learning model for BAV classification that is trained using unlabeled MRI data and weak supervision as a surrogate for the intensive workload of trained data sets. The data set of 14,328 UK Biobank subjects and 412 "gold-standard" patients appears to be of sufficient size, but of these only 17 had BAV, identified by billing codes and by review by cardiologist. Incidentally, I think you looked at 423 patients, not 412. My concern with this small number is that it barely covers the number of anatomical BAV variants and may well have failed to include a "true" bicuspid (Sievers type 0), a L-N type 1, or type 2 sub-types.

Later you describe a training set of 50 BAV and 50 TAV. It was not clear to me how this training set related to the 17 BAV and 396 TAV gold standard annotations you describe earlier.

To my eye, the real validation is to definitive anatomical findings, not to cardiovascular outcomes. Using this fairly weak surrogate of anatomy does not adequately define the accuracy of the methods you describe. It would be easy to argue that the performance of a cardiac MRI was not randomly assigned across the UK Biobank population, or that the roughly two-fold increased risk of outcomes might be five-fold if the algorithm was entirely accurate.

Next, in Table 1, you describe 106 "BASELINE" patients; is this not the same as the 100 training patients described in the paragraphs above, in the paper?

Based on the text and Figure 5, you describe 570 / (570+8660) patients having BAV; a population rate of about 6.2%. This is some 4-5 fold higher than observed in echocardiographic studies, implying a very large false positive rate or profound bias in patient ascertainment. I strongly suggest manually reviewing all 570 of these patients' MRIs to validate the algorithm.

Reviewer #2 (Remarks to the Author):

I am reviewing this paper from a machine learning perspective.

The paper proposes to use weak supervision (in particular the Snorkel system) for a medical image classification task. The motivation for this methodology is sound: labeling data is too costly but there is an abundance of unlabeled data. The authors have found a set of simple features that can be extracted from the images in an unsupervised way. This enables a set of weak labeling functions to provide the noisy supervision signal. A learned generative model tries to correct for the correlations between these noisy supervision signals.

I very much like this general approach to machine learning, it makes perfect sense. That being said, the approach is quite well-known and there is not much novelty on the machine learning side. I don't think this is a problem if on the medical side, the paper is proposing something novel and significant. Then I think it should be accepted, even as is. It can be important to expose various application fields to this weak supervision methodology.

My only concern on the machine learning side is that the validation set is relatively small (as it always will be in a setting such as this one), yet there are many hyperparameters to tune, and even more concerning, the soft labeling process at the bottom of page 8 is very specific and complex. I find it hard to believe this soft labeling process was properly tuned on a handful of validation examples, and there is some risk it could be overfitting on the test set.

Why not use some long-term outcomes or other side information as a related task to get some supervision from?

The sentence "infer higher order dependency structure between labeling functions based on their interactions with primitives" is not understandable without more explanation.

The factor graph in Fig 2 has a factor between lambda 1 and 4 (also 3 and 5). These factors don't have any effect on $P(Y|\text{Lambda})$. It's a detail, but I wonder if you could explain why they are included in the generative model.

The term generative model is used in two different meanings in different communities. Some communities take it to mean a joint distribution, as does this papers. Others take it to mean a joint distribution that is efficient to sample from, like a directed graphical model. In that latter community, a factor graph is not a generative model.

I appreciate very much the code being available.

Reviewer #1

Identification and characterization of bicuspid aortic valve anatomy is an human-intensive task with considerable inter-observer variability and about a 5-10% failure to accurately define the valve. The problem is notable for the several anatomical variants and difficulty identifying partial commissural fusion. The problem is especially difficult using gated raw-image cardiac MRI, but the value of avoiding ionizing radiation in long-term management of these patients is important. These imaging problems is a hindrance to clinical diagnosis, high-quality epidemiologic studies and accurate phenotyping for high-N genetic studies.

We thank the reviewer for their positive comments.

The authors present a deep learning model for BAV classification that is trained using unlabeled MRI data and weak supervision as a surrogate for the intensive workload of trained data sets. The data set of 14,328 UK Biobank subjects and 412 “gold-standard” patients appears to be of sufficient size, but of these only 17 had BAV, identified by billing codes and by review by cardiologist. Incidentally, I think you looked at 423 patients, not 412. My concern with this small number is that it barely covers the number of anatomical BAV variants and may well have failed to include a “true” bicuspid (Sievers type 0), a L-N type 1, or type 2 sub-types.

The reviewer is entirely correct that the small number of patients surveyed is not sufficient to sample all of the anatomical sub-classes of bicuspid aortic valve. We have more clearly noted this limitation in the discussion. The patient count discrepancy was due to a typo and has been corrected.

Later you describe a training set of 50 BAV and 50 TAV. It was not clear to me how this training set related to the 17 BAV and 396 TAV gold standard annotations you describe earlier.

We apologize for the lack of clarity. The 50 BAV/50 TAV are from our weakly labeled MRI set. These samples (and all scale-up intervals 50 - 4000) are generated using our label model and are used for training the CNN-LSTM. This set of subjects is disjoint from the 412 subjects used for gold standard annotations. We have changed the text to more clearly represent our experimental approach.

To my eye, the real validation is to definitive anatomical findings, not to cardiovascular outcomes. Using this fairly weak surrogate of anatomy does not adequately define the accuracy of the methods you describe.

We thank the reviewer for the their comments. We have now hand-validated a subset of the classified MRIs to more clearly understand the accuracies and weaknesses of our algorithm. However we believe that the clinical outcomes data serves as additional validation of the algorithm--rarely are clinical outcomes data (completely separate from data used to build a model) available in technical descriptions of ML in healthcare imaging applications.

It would be easy to argue that the performance of a cardiac MRI was not randomly assigned across the UK Biobank population, or that the roughly two-fold increased risk of outcomes might be five-fold if the algorithm was entirely accurate.

For the MRI portion of the UKB study, individuals with metal implants, recent surgery, or other health conditions precluding a prolonged time in the magnet, were excluded from participating (PMID 27643430 (Miller et al. 2016), UKB imaging information leaflet <https://bit.ly/2EqXZMw>). So the reviewer is correct that participating individuals with MRI data are healthier than the rest of the UKB cohort by every standard measure (illustrated in the table below). Due to the selection bias for healthy individuals and classification error, the magnitude of the risk estimates may not be comparable to values that would be expected in the general population at a fine scale. Despite a selection bias toward younger and healthier individuals, the clinical data validate the direction and overall magnitude of risk for MACE expected in a group of individuals with aortic valve pathology, using data completely independent of the process of visual classification. We have added additional discussion and clarification of this point throughout the manuscript.

	MRI No (n=490,650)	MRI Yes (n=9,339)	p-value
Age_baseline (sd)	56.4 (8.1)	54.9 (7.5)	<0.001
Male Sex (%)	223,181 (45.5)	4,568 (48.9)	<0.001
Townsend Index (sd)	-1.3 (3.1)	-2.0 (2.6)	<0.001
Smoking (%)			
Never	266,653 (54.7)	5,650 (60.6)	
Current	52,097 (10.7)	599 (6.4)	
Previous	168,991 (34.6)	3,068 (32.9)	
BMI, mean (sd)	27.5 (4.8)	26.7 (4.2)	<0.001
Hypertension (%)	269,726 (55.3)	4,362 (46.5)	<0.001

Hyperlipidemia (%)	92,012 (18.9)	1,335 (14.9)	<0.001
Diabetes (%)	18,640 (3.8)	204 (2.2)	<0.001

Next, in Table 1, you describe 106 “BASELINE” patients; is this not the same as the 100 training patients described in the paragraphs above, in the paper?

We thank the reviewer for the opportunity to clarify our approach. The BASELINE model is trained on the 106 hand-labeled MRIs comprising our gold annotated development set, which was used to develop labeling functions. The 100 trained examples discussed in the preceding paragraph are weakly labeled MRIs generated by our label model. We have attempted to alter the text to more clearly represent our experimental approach.

Based on the text and Figure 5, you describe 570 / (570+8660) patients having BAV; a population rate of about 6.2%. This is some 4-5 fold higher than observed in echocardiographic studies, implying a very large false positive rate or profound bias in patient ascertainment. I strongly suggest manually reviewing all 570 of these patients' MRIs to validate the algorithm.

We thank the reviewer for their comments. We have now hand-validated a large subset (n=100) of the classified MRIs to more clearly understand the accuracies and weaknesses of our algorithm. In total of 75% of the reviewed images display a clear pathology of the aortic valve including 28% with BAV, 37% with aortic stenosis, 11% with aortic insufficiency, and 16% displaying one or more tethered or thickened leaflets of the aortic valve. We have added these key data to the manuscript and additional discussion therein focusing on the fact that the algorithm specifically detects BAV in addition to detecting other common pathologies of the aortic valve.

Reviewer #2

I am reviewing this paper from a machine learning perspective.

The paper proposes to use weak supervision (in particular the Snorkel system) for a medical image classification task. The motivation for this methodology is sound: labeling data is too costly but there is an abundance of unlabeled data. The authors have found a set of simple features that can be extracted from the images in an unsupervised way. This enables a set of weak labeling functions to provide the noisy supervision signal. A learned generative model tries to correct for the correlations between these noisy supervision signals.

We thank the reviewer for their comments.

I very much like this general approach to machine learning, it makes perfect sense. That being said, the approach is quite well-known and there is not much novelty on the machine learning side. I don't think this is a problem if on the medical side, the paper is proposing something novel and significant. Then I think it should be accepted, even as is. It can be important to expose various application fields to this weak supervision methodology.

We appreciate the reviewer's perspective and indeed would position our efforts as employing these principles on real-world medical data, with issues such as severe class imbalance and the noise and messiness of large-scale, prospectively obtained study data.

My only concern on the machine learning side is that the validation set is relatively small (as it always will be in a setting such as this one), yet there are many hyperparameters to tune, and even more concerning, the soft labeling process at the bottom of page 8 is very specific and complex. I find it hard to believe this soft labeling process was properly tuned on a handful of validation examples, and there is some risk it could be overfitting on the test set.

We thank the reviewer for their comment and sharing their concerns. We note the final test set was withheld during all points of model development. We agree that labeling heuristics are likely biased to the small validation set used for soft label development. We demonstrate that despite the risk of overfitting, the training labels assigned by these labeling functions and the deep learning model are able to generalize to perform well on the unseen test set. In our post-hoc analysis of the prediction set, our models generalize enough to increase the number of BAV subjects detected by an order of magnitude (2% to 28%). Bias manifests as failing to capture specific BAV subtypes, as noted by Reviewer 1. In these cases, we have no examples to guide labeling function development.

Why not use some long-term outcomes or other side information as a related task to get some supervision from?

We deliberately assumed a conservative setting for defining weak supervision, i.e., restricting ourselves to image features and coarse demographic variables (age, sex). We agree that incorporating side information from ICD9/10 codes, other MRI views, etc. to leverage data on long-term outcomes and confounding pathologies is an exciting area for future model improvement. However, we felt this deserved more detailed treatment than was possible in this manuscript and instead focused on performance given image data alone.

The sentence "infer higher order dependency structure between labeling functions based on their interactions with primitives" is not understandable without more explanation.

We thank the reviewer for the opportunity to clarify this statement. Labeling functions in our setting interact with a small set of primitives, e.g., area and perimeter, and have higher order dependency structure. For example, a labeling function defined using the ratio of area and perimeter (LF 5) has dependencies with separate labeling functions for area (LF 1) and perimeter (LF 3). By expressing supervision using a small space of primitives, we can rely on the method described in (Varma et al. 2017) to statically analyze source code and automatically infer complex dependencies among labeling functions based on which primitives the functions operate over. We have changed the manuscript text to describe this algorithmic approach more clearly.

The factor graph in Fig 2 has a factor between lambda 1 and 4 (also 3 and 5). These factors don't have any effect on $P(Y|\Lambda)$. It's a detail, but I wonder if you could explain why they are included in the generative model.

We thank the reviewer for their comment. For exposition purposes, we formally described the conditionally independent data programming model, however our pipeline uses a formulation that supports arbitrary dependencies between labeling functions. We have updated the manuscript to describe this dependency-based model.

The term generative model is used in two different meanings in different communities. Some communities take it to mean a joint distribution, as does this paper. Others take it to mean a joint distribution that is efficient to sample from, like a directed graphical model. In that latter community, a factor graph is not a generative model.

We thank the reviewer for their comment-- we have updated the manuscript to clarify that we refer to a factor-graph based label model that can learn parameters that represent the accuracy and dependencies of the labeling functions and generate probabilistic training labels for the unlabeled data.

I appreciate very much the code being available.

We thank the reviewer for their noticing our effort to engage in open and collaborative science.

References

- Miller, Karla L., Fidel Alfaro-Almagro, Neal K. Bangerter, David L. Thomas, Essa Yacoub, Junqian Xu, Andreas J. Bartsch, et al. 2016. "Multimodal Population Brain Imaging in the UK Biobank Prospective Epidemiological Study." *Nature Neuroscience* 19 (11): 1523–36.
- Varma, Paroma, Bryan He, Payal Bajaj, Imon Banerjee, Nishith Khandwala, Daniel L. Rubin, and Christopher Ré. 2017. "Inferring Generative Model Structure with Static Analysis." *Advances in Neural Information Processing Systems* 30 (December): 239–49.

REVIEWERS' COMMENTS:

Reviewer #1 (Remarks to the Author):

Thank you for the extra work and explanatory revisions. The paper is much improved with much more realistic findings.

You have done a good job discussing the implications of the work in the Discussion, but I think you need to be more explanatory in the title and abstract. Dealing with the Title: BAV and structural and functional aortic valve disease isn't that rare. You observed it about 6% of the time.

In the Abstract you stated "We developed a weakly supervised deep learning model for Bicuspid Aortic Valve (BAV) classification using up to 4,000 unlabeled cardiac MRI sequences. Instead of requiring curated, hand labeled training data, weak supervision relies on noisy heuristics defined by domain experts to programmatically generate large-scale, imperfect training labels. For BAV classification, training models using these imperfect labels substantially outperformed a traditional supervised model trained on hand-labeled MRIs." I think you need to more accurately state your findings.

The crux of the paper now has to do with identifying any form of aortic valve disease, not just BAV. You do have some BAVs in the dataset, but the rest are people with apparent tricuspid aortic valve disease. So rather than showing weakly-supervised learning from cardiac MRIs can identify BAV patients, you have shown that weakly-supervised learning can identify aortic valve disease. That's not a bad finding, but it has to be realistically presented.

Reviewer #2 (Remarks to the Author):

I have checked the response to reviewers and modifications to the paper, and am happy with the changes. Since I was already in favor of accepting the paper, and the paper was clearly further improved in this second round, I am again recommending acceptance.

REVIEWERS' COMMENTS:

Reviewer #1 (Remarks to the Author):

Thank you for the extra work and explanatory revisions. The paper is much improved with much more realistic findings.

You have done a good job discussing the implications of the work in the Discussion, but I think you need to be more explanatory in the title and abstract. Dealing with the Title: BAV and structural and functional aortic valve disease isn't that rare. You observed it about 6% of the time.

In the Abstract you stated "We developed a weakly supervised deep learning model for Bicuspid Aortic Valve (BAV) classification using up to 4,000 unlabeled cardiac MRI sequences. Instead of requiring curated, hand labeled training data, weak supervision relies on noisy heuristics defined by domain experts to programmatically generate large-scale, imperfect training labels. For BAV classification, training models using these imperfect labels substantially outperformed a traditional supervised model trained on hand-labeled MRIs." I think you need to more accurately state your findings.

The crux of the paper now has to do with identifying any form of aortic valve disease, not just BAV. You do have some BAVs in the dataset, but the rest are people with apparent tricuspid aortic valve disease. So rather than showing weakly-supervised learning from cardiac MRIs can identify BAV patients, you have shown that weakly-supervised learning can identify aortic valve disease. That's not a bad finding, but it has to be realistically presented.

We thank Reviewer 1 for their comments and suggestions. We have updated the abstract, introduction, and discussion text to more accurately reflect that our models identify general aortic valve disease, inclusive of BAV.

Reviewer #2 (Remarks to the Author):

I have checked the response to reviewers and modifications to the paper, and am happy with the changes. Since I was already in favor of accepting the paper, and the paper was clearly further improved in this second round, I am again recommending acceptance.

We thank Reviewer 2 for their support.